# Distributed Event-Triggered Synchronization for Complex Cyber–Physical Networks under DoS Attacks

**Xiaojie Huang [1,2], Yunxia Xia [3,\*] and Da-Wei Ding [1,4,\*]**

[1] School of Automation and Electrical Engineering, University of Science and Technology Beijing, Beijing 100083, China

[2] Shunde Innovation School, University of Science and Technology Beijing, Foshan 528399, China

[3] Institute of Optics and Electronics, Chinese Academy of Sciences, Chengdu 610209, China

[4] Key Laboratory of Knowledge Automation for Industrial Processes of Ministry of Education, Beijing 100083, China

\* Correspondence: xyxhaha@163.com (Y.X.); ddaweiauto@163.com (D.-W.D.)

**Abstract:** With the continuous development of the networked society, the ability of cyber attackers is becoming increasingly intelligent, posing a huge threat to complex cyber–physical networks (CCPNs). Therefore, how to design a security strategy for CCPNs under attack has become an urgent problem to be solved, which promotes our work. The problem of the distributed event-triggered synchronization of CCPNs in the presence of denial-of-service (DoS) attacks is investigated in this paper. Firstly, a distributed event-triggered controller is designed such that all nodes of networks are synchronized without DoS attacks by relieving the communication occupancy rate of limited bandwidths. Meanwhile, Zeno and singular triggering behaviors are excluded to illustrate the effectiveness of the proposed event-triggered strategy. Secondly, in view of the continuous switching of CCPNs topologies caused by DoS attacks, an event-triggered control (ETC) strategy is proposed to ensure the synchronization of CCPNs under DoS attacks. Meanwhile, the frequency and duration of tolerable DoS attacks that can ensure the stability of the systems are calculated. Finally, two examples are given to illustrate the effectiveness of the proposed method.

**Keywords:** complex cyber–physical networks; distributed controller; ETC strategy; synchronization; DoS attacks

## 1. Introduction

With the development of complex dynamical network theory and its application in recent years, complex dynamical networks (CDNs) have become a hot research topic in many fields [1]. CDNs are composed of many nodes and edges, which are used to represent different individuals and connections between individuals. In particular, it is of great importance to investigate the connection of edges and the communication of nodes. In recent years, CDNs have been widely used to simulate many large-scale practical complex projects [2–6]. Many topics on CDNs have been discussed, including synchronization, propagation dynamics and fault diagnosis, control and network model [7–13], etc. On the other hand, complex cyber–physical networks (CCPNs) as a special type of CDNs have received extensive attention. CCPNs are multi-dimensional complex systems consisting of computing network and physical environment, which realize the integrated design of computing communication and physical systems. Therefore, it has a wide application prospect, such as medical science, transportation networks, electric power grid and manufacturing [14–20]. Many interesting issues on CCPNs have been intensively investigated in the literature. Specially, the synchronization problem of CCPNs has been discussed in [21–25].

Due to high dependence on networks, CCPNs are faced with many cyber attack threats. Typical cyber attacks main contain false data injection (FDI) attacks and denial-of-service (DoS) attacks [15,21,26,27]. Attackers use cyber attacks to interfere with the

information space of the system or destroy data, which can affect the physical space of the system and even destroy the whole system. In particular, DoS attacks can block the transmission links of networks such that the transmission information received by actuators and sensors can be corrupted, thereby affecting the system performance. Many research studies have been reported on the security of cyber–physical systems (CPSs) under DoS attacks. For example, the event-triggered secure control for multi-agent systems with DoS attacks was investigated in [28–33], in which a multi-robot model was proposed in [30], keeping all six robots in a line and achieving the consensus of their position and velocity under the designed event-triggered control (ETC) strategy. The distributed controller was designed for CPSs with communication delays against attacks in [34–36]. Unfortunately, the above existing results only discussed the convergence of the systems based on the controller and did not investigate the strong coupling between individuals within the systems. That is, the security synchronization control has not been fully investigated for CCPNs, which have highly coupled nodes and intricate communication links, making it difficult to solve. This motivates the present work.

In this paper, a distributed event-triggered control strategy is proposed for CCPNs to update the state transmission under DoS attacks, which relieves the communication occupancy rate of limited bandwidths and achieves synchronization of all the nodes. The following are the contributions of this paper:

(1) A distributed controller is designed to achieve all nodes of CCPNs are synchronized under DoS attacks, which uses two combinational measurements of isolated node and follower nodes.

(2) An ETC strategy is proposed to save communication resources under DoS attacks. Different from [25,28,30,36], triggering behavior and Zeno behavior are both excluded in this paper, which avoids continuous triggering in each verification cycle and guarantees that certain data are still transmitted after a successful transmission attempt.

(3) A resilient event-triggering mechanism is constructed to tolerate packet loss and topologies change under DoS attacks. Attack duration and frequency can be calculated to describe attack behavior based on piecewise Lyapunov functions.

**Notation:** $\mathcal{L}$ represents the Laplacian matrix of graph $\mathcal{G} = \{v, \varepsilon, \mathcal{A}\}$. $\Gamma(P) = \sqrt{\theta(PP)}$, and $\theta(P)$ represents the eigenvalue of matrix $P$.

## 2. Problem Formulation and Preliminaries

The CCPNs with $N$ nodes are described as follows:

$$\dot{x}_i(t) = Ax_i(t) + \sum_{j=1}^{N} \mathfrak{g}_{ij}\Gamma x_j(t) + Bu_i(t) \tag{1}$$

where $x_i(t) = (x_{i1}(t), \ x_{i2}(t), \dots, \ x_{in}(t))^T \in \Re^n$ is the state vector of node $i$, and $u_i \in \Re^q$ is the control input of node $i$. $A \in \Re^{n \times n}$, $B \in \Re^{n \times q}$ are constant matrices. $\Gamma$ and $G = (\mathfrak{g}_{kl})_{N \times N}$ are inner-coupling matrix and outer-coupling matrix, respectively. $\mathfrak{g}_{kl} = \mathfrak{g}_{lk} = 1$, if node $k$ and node $l$ $(k \neq l)$ are connected; otherwise, $\mathfrak{g}_{kl} = \mathfrak{g}_{lk} = 0$ $(k \neq l)$. And $\sum_{l=1, l \neq k}^{N} \mathfrak{g}_{kl} = -\mathfrak{g}_{kk}$.

Let $x_{N+1}(t)$ be the isolated node whose dynamics are given as follows: $\dot{x}_{N+1}(t) = Ax_{N+1}(t)$. Denote the tracking error as

$$\delta_i(t) = x_i(t) - x_{N+1}(t) \tag{2}$$

For node $i$, designing a distributed ETC protocol is as follows:

$$u_i(t) = K\hat{\xi}_i\left(t_k^i\right), \quad t \in \left[t_k^i, t_{k+1}^i\right) \tag{3}$$

$$\hat{\xi}_i\left(t_k^i\right) = \sum_{j=1}^{N} a_{ij}\left(x_j(t_k^i) - x_i\left(t_k^i\right)\right) + b_i\left(x_{N+1}\left(t_k^i\right) - x_i\left(t_k^i\right)\right) \tag{4}$$

where $\{t_k^i\}_{k\in\mathcal{N}}$ denotes the event-triggering time sequence for node $i$. If node $i$ can obtain information from the isolated node, then $b_i = 1$; otherwise $b_i = 0$.

**Definition 1.** *All nodes of CCPNs (1) are said to be synchronized if the following condition is satisfied:*

$$\lim_{t\to\infty}[\|x_i(t) - x_{N+1}(t)\|] = 0 \tag{5}$$

The proposed attack model in this paper is a time-constrained DoS attack, which mainly attacks the nodes and communication links of the systems in the physical layer. The attack behavior is described below. $J_s = \left[j_s^{ij}, j_s^{ij} + \Delta_s^{ij}\right], s \in \mathcal{N}$ defines the $s$-th DoS time interval, where the DoS attack begins at $j_s^{ij}$ and $\Delta_s^{ij} > 0$ represents the attack duration of the $s$-th attack. Then, one can obtain

$$\Theta_a(t_1, t_2) = [t_1, t_2] \cap \cup \Theta_a^{(i,j)}(t_1, t_2) \tag{6}$$

and

$$\Theta_s(t_1, t_2) = [t_1, t_2] \backslash \Theta_a(t_1, t_2) \tag{7}$$

where $\Theta_a^{(i,j)}(t_1, t_2) = [t_1, t_2] \cap \overset{\infty}{\underset{s=1}{\cup}} J_s^{(i,j)}$ is a set in which the DoS attacks are active. $\Theta_a(t_1, t_2)$ and $\Theta_s(t_1, t_2)$ denote the sets that the DoS attacks are active and dormant, respectively.

Figure 1 is a demonstration of a CCPN with six nodes under DoS attacks, in which node 6 is the isolated node. As shown in Figure 1, the malicious attacks are active at $t_0$, and the systems are repaired at $\tilde{t}_0$. The system transmission channel can be restored at time $t_1$. That is, the communication graphs are discontinuous between $\tilde{t}_0$ to $t_1$. The communication topologies of networks can be restored to the initial state at $t_2$.

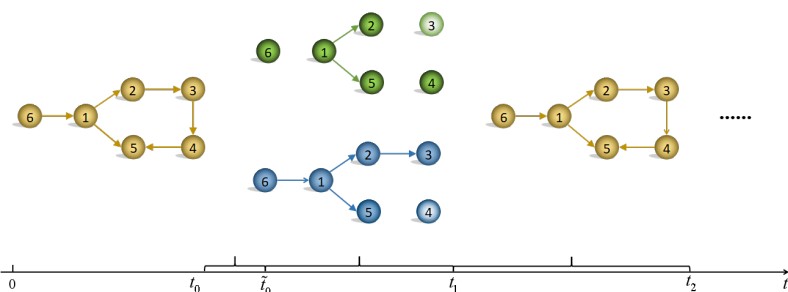

**Figure 1.** The evolution of DoS attacks.

**Remark 1.** *DoS attacks destroy the nodes or communication edges of networks such that communication links fail or nodes do not work normally. Meanwhile, without loss of generality, suppose that at least one communication link or one node in dynamic network systems is attacked at time $t_{2l}^{ij}$ for $l = 0, 1, \ldots, N$. $t_{2l+1}^{ij}$ is the time instant when the attack has been completely restored and the entirely topology network has been restored. It is worth noticing that the CCPNs are paralyzed during $t \in \left[t_{2l}^{ij}, t_{2l+1}^{ij}\right)$, and the CCPNs work well during $t \in \left(t_{2l+1}^{ij}, t_{2(l+1)}^{ij}\right]$.*

Define $\Lambda_*$ as the continuous impact of DoS attacks to ensure that the sampling and transmission of nodes are successful. Then the time interval of $s$-th attack is re-denoted as follows:

$$\tilde{J}_s^{(i,j)} = \left[j_s^{ij}, j_s^{ij} + \Delta_s^{ij} + \Lambda_*\right], s \in \mathcal{N} \tag{8}$$

$$\tilde{\Theta}_a(t_1, t_2) = [t_1, t_2] \cap \cup \tilde{J}_s^{(i,j)} \tag{9}$$

and

$$\tilde{\Theta}_s(t_1, t_2) = [t_1, t_2] \backslash \tilde{\Theta}_a(t_1, t_2) \tag{10}$$

we propose the following equations to characterize the case that the topologies are still connected under the DoS attacks

$$\Theta_c(t_1, t_2) = \left\{ t \in (t_1, t_2) \Big| \theta_{\min}\left( \bar{H}_{\sigma(t)} \right) > 0 \right\} \tag{11}$$

where $\sigma(t)$ is a switching signal and topologies are supposed to be completely normal when $\sigma(t) = 0$. Finally, the characteristics of DoS attacks are rewritten as

$$\tilde{\Theta}_{a'}(t_1, t_2) = \tilde{\Theta}_a(t_1, t_2) \backslash \Theta_c(t_1, t_2) \tag{12}$$

and

$$\tilde{\Theta}_{s'}(t_1, t_2) = [t_1, t_2] \backslash \tilde{\Theta}_{a'}(t_1, t_2) \tag{13}$$

**Definition 2** ([30] (Attack Frequency)). *Define $F_{a_{(T_1, T_2)}}$ as the frequency of DoS attacks, satisfying*

$$F_{a_{(T_1, T_2)}} = \frac{N_{a_{(T_1, T_2)}}}{T_2 - T_1}$$

*where $N_{a_{(T_1, T_2)}}$ is the number of DoS attacks occurring over $[T_1, T_2)$, and $T_2 > T_1 > t_0$.*

**Definition 3** ([30] (Attack Duration)). *Define $T_{a_{(T_1, T_2)}}$ as the total time of DoS attacks over $[T_1, T_2)$, satisfying*

$$T_0 + \frac{T_2 - T_1}{\tau_a} \geq T_{a_{(T_1, T_2)}}$$

*where $T_0 > 0, \tau_a > 0, T_2 > T_1 > t_0$.*

**Remark 2.** *It is noted that the transmission channel between nodes may be clogged instead of being removed under DoS attacks. On one hand, to exclude the communication topologies that are still normal under DoS attacks, (11) and (12) are designed to ensure that communication topologies of CCPNs are destroyed under DoS attacks. On the other hand, different from the DoS attack interval defined in [30], the attack interval is designed as (8) in this paper, which includes diversified DoS attack opponents that destroy communication links or nodes or both. In other words, the directed spanning tree of communication networks becomes disconnected under DoS attacks.*

In this paper, we design an event-triggered distributed controller to guarantee that all nodes of CCPNs are synchronized. The synchronization error can be defined as $e_i(t) = x_i\left(t_k^i\right) - x_i(t)$. The virtual DoS period is divided into two parts: $\left[t_{2l}^{ij}, t_{2l+1}^{ij}\right) \cup \left(t_{2l+1}^{ij}, t_{2(t+1)}^{ij}\right]$. The topologies are paralyzed by attacks of adversaries during $t \in \left[t_{2l}^{ij}, t_{2l+1}^{ij}\right)$ and the communication topologies are assumed to be complete and $\sigma(t) = 0$ during $t \in \left(t_{2l+1}^{ij}, t_{2(t+1)}^{ij}\right]$.

Then, the control law for node $i$ is designed as

$$u_i(t) = \begin{cases} F\hat{\xi}_{i1}(t), & t \in \left[t_{2l}^{ij}, t_{2l+1}^{ij}\right) \\ K\hat{\xi}_{i2}(t), & t \in \left[t_{2l+1}^{ij}, t_{2(l+1)}^{ij}\right) \end{cases} \tag{14}$$

where

$$\hat{\xi}_{i1}\left(t_k^i\right) = \sum_{j=1}^{N} a_{ij}^{\sigma(t)} \left( x_j\left(t_k^i\right) - x_i\left(t_k^i\right) \right) + b_i^{\sigma(t)} \left( x_{N+1}\left(t_k^i\right) - x_i\left(t_k^i\right) \right) \tag{15}$$

$$\hat{\xi}_{i2}\left(t_k^i\right) = \sum_{j=1}^{N} a_{ij}^0 \left(x_j\left(t_k^i\right) - x_i\left(t_k^i\right)\right) + b_i^0 \left(x_{N+1}\left(t_k^i\right) - x_i\left(t_k^i\right)\right) \tag{16}$$

and $\{t_k^i\}$ is the updated state transmission sequence for each node $i$ based on the ETC strategy. $x_i(t_k^i) = \hat{x}_i(t)$ is the latest transmission state, and $x_{N+1}(t_k^i) = x_{N+1}(t)$. $\{t_k^i\}$ of each node satisfies the following triggering condition:

$$\mathfrak{F}\left(e_i(t), \hat{\xi}_i\left(t_k^i\right)\right) = 0 \tag{17}$$

**Remark 3.** *An event-triggered transmission strategy is proposed to save communication resources in this paper. Equation (17) states that the state-triggering time of each node is independent and not affected by other nodes. In addition, the event-triggering condition in [30] only satisfies a specified inequality, and the selection of parameters has a significant impact on the stability of the systems. On the contrary, the event-triggering condition (17) obtained on the basis of the system stability in this paper can better ensure the performance of systems. For each node i, if $t_k^i$ satisfies the update condition (17), the $\hat{x}_i(t)$ will update at event-triggering instant $t_k^i$.*

Then, the error systems can be written as

$$
\begin{aligned}
&t \in \left[t_{2l}^{ij}, t_{2l+1}^{ij}\right), \\
&\dot{\delta}_i(t) = A\delta_i(t) + BF\left[\sum_{j=1}^{N} a_{ij}^{\sigma(t)}\left(x_j\left(t_k^i\right) - x_i\left(t_k^i\right)\right) + b_i^{\sigma(t)}\left(x_{N+1}\left(t_k^i\right) - x_i\left(t_k^i\right)\right)\right]
\end{aligned}
\tag{18}
$$

and

$$
\begin{aligned}
&t \in \left[t_{2l+1}^{ij}, t_{2(l+1)}^{ij}\right), \\
&\dot{\delta}_i(t) = A\delta_i(t) + BK\left[\sum_{j=1}^{N} a_{ij}^0\left(x_j\left(t_k^i\right) - x_i\left(t_k^i\right)\right) + b_i^0\left(x_{N+1}\left(t_k^i\right) - x_i\left(t_k^i\right)\right)\right]
\end{aligned}
\tag{19}
$$

Let $\dot{x}_{N+1}\left(t_k^i\right) = \dot{x}_{N+1}(t)$, and from the system (18) and (19), we have

$$
\dot{\delta}(t) = 
\begin{cases}
\left(I_N \otimes A + G \otimes \Gamma - \bar{H}_{\sigma(t)} \otimes BF\right)\delta(t) - \left(\bar{H}_{\sigma(t)} \otimes BF\right)e(t), t \in \left[t_{2l}^{ij}, t_{2l+1}^{ij}\right) \\
(I_N \otimes A + G \otimes \Gamma - H \otimes BK)\delta(t) - (H \otimes BK)e(t), t \in \left(t_{2l+1}^{ij}, t_{2(l+1)}^{ij}\right]
\end{cases}
\tag{20}
$$

where $\delta(t) = col(\delta_1(t), \delta_2(t), \ldots, \delta_N(t))$, $e(t) = col(e_1(t), e_2(t), \ldots, e_N(t))$, $H = \mathcal{L} + \mathcal{B}$, $\bar{H}_{\sigma(t)} = \mathcal{L}_{\sigma(t)} + \mathcal{B}_{\sigma(t)}$, $\mathcal{B} = diag(b_1, b_2, \ldots, b_N)$, $\mathcal{B}_{\sigma(t)} = diag\left(b_1^{\sigma(t)}, b_2^{\sigma(t)}, \ldots, b_N^{\sigma(t)}\right)$.

The goal of this paper is to design an event-triggered distributed control law (3) such that CCPNs (1) under DoS attacks are stable and the normal transmission of information is guaranteed.

The following lemma is needed in the sequel.

**Lemma 1** ([25]). *For positive definite matrix $M \in \Re^{n \times n}$ and symmetric matrix $N \in \Re^{n \times n}$, the following inequalities hold*

$$\theta_{\min}\left(M^{-1}N\right)x^T Tx \le x^T Nx \le \theta_{\max}\left(M^{-1}N\right)x^T Mx$$

$$\theta_{\min}(N)x^T x \le x^T Nx \le \theta_{\max}(N)x^T x$$

## 3. Main Results

We propose an ETC data update strategy to guarantee that all nodes of CCPNs are synchronized in this section. Then, we solve the topologies change caused by DoS attacks by a distributed controller.

**Theorem 1.** *The all nodes of CCPNs* (1) *without DoS attacks can be synchronized based on the event-triggered distributed controller* (14) *with* $K = B^T P$ *satisfying*

$$A^T P + PA - 2\varepsilon_{\min} P\Gamma - 2\varepsilon_{\min} PBB^T P + \varepsilon_{\min} I_N < 0 \tag{21}$$

*and the triggering function*

$$\mathfrak{F}(e_i, \hat{\xi}_{i2}) = \|e_i\| - \sqrt{\frac{\gamma_i(q_1 - q_2\rho)}{q_2\rho^{-1} + q_3}} \|\hat{\xi}_{i2}\| \tag{22}$$

*Moreover, the event-triggering time is defined as*

$$\tau_k^i = \inf\left\{ t > t_k^i \middle| \mathfrak{F}(e_i, \hat{\xi}_{i2}) = 0 \right\}, \qquad k = 1, 2, \dots \tag{23}$$

*where* $\rho > 0$, $0 < \gamma_i < 1$ *and* $q_1 - q_2\rho > 0$, $\varepsilon_{\min} = \max\{\theta(G), \theta(H)\}$,
$0 < \varepsilon_1 < \varepsilon_{\min}$, $\varepsilon_{\min} = \varepsilon_1 + \varepsilon_2$, $q_1 = \min\{\varepsilon_1 \Gamma_{\min}(H^{-2})\}$,
$q_2 = \max\{\Gamma_{\max}\{H^{-1} \otimes 2\varepsilon_1 I_N - I_N \otimes 2PBB^T P\}\} > 0$,
$q_3 = \max\{\Gamma_{\max}\{I_N \otimes 2PBB^T P\} - \varepsilon_1\} > 0$.

**Proof.** The following Lyapunov function is constructed for systems (19)

$$V_a(t) = \delta^T (I_N \otimes P)\delta \tag{24}$$

From (16), one can obtain

$$\begin{aligned} \hat{\xi}_{i2}(t) &= \sum_{j=1}^N a_{ij}^0 \left(x_j(t_k^i) - x_i(t_k^i)\right) + b_i^0 \left(x_{N+1}(t_k^i) - x_i(t_k^i)\right) \\ &= -(H \otimes I_N)\left(x_i(t_k^i) - x_{N+1}(t)\right) \end{aligned} \tag{25}$$

and

$$\delta_i(t) = x_i(t_k^i) - e_i(t) - x_{N+1}(t) \tag{26}$$

Combining (25) and (26), we have

$$\delta = -\left(H^{-1} \otimes I_N\right)\hat{\xi}_2 - e \tag{27}$$

The time derivatives of (24) yield

$$\begin{aligned} \dot{V}_a(t) &= 2\delta^T (I_N \otimes P)\dot{\delta} \\ &= 2\delta^T (I_N \otimes P)[(I_N \otimes A + G \otimes \Gamma - H \otimes BK)\delta - (H \otimes BK)e] \\ &= \delta^T (A^T P + PA + G \otimes 2P\Gamma - H \otimes 2PBB^T P)\delta - \delta^T (H \otimes 2PBB^T P)e \\ &\leq -\varepsilon_{\min}\delta^T \delta - \delta^T (H \otimes 2PBB^T P)e \\ &\leq -(\varepsilon_1 + \varepsilon_2)\left\{\hat{\xi}_2^T (H^{-2} \otimes I_N)\hat{\xi}_2 + 2\hat{\xi}_2^T (H^{-1} \otimes I_N)e + e^T e\right\} \\ &\quad - e^T (I_N \otimes 2PBB^T P)\hat{\xi}_2 + e^T (I_N \otimes 2PBB^T P)e \\ &\leq -\varepsilon_1 \hat{\xi}_2^T (H^{-2} \otimes I_N)\hat{\xi}_2 + 2\varepsilon_1 \hat{\xi}_2^T (H^{-1} \otimes I_N)e - \varepsilon_1 e^T e \\ &\quad - e^T (I_N \otimes 2PBB^T P)\hat{\xi}_2 + e^T (I_N \otimes 2PBB^T P)e - \varepsilon_2 \delta^T \delta \end{aligned} \tag{28}$$

Combine with $M^T N \leq \|M\|\|N\|$, $M^T N \leq \frac{1}{2h}\|M\|^2 + \frac{h}{2}\|N\|^2$, then we have

$$\begin{aligned} -\varepsilon_1 \hat{\xi}_2^T (H^{-2} \otimes I_N)\hat{\xi}_2 &\leq -q_1 \|\hat{\xi}_2\|^2, \quad -\varepsilon_1 e^T e + e^T (I_N \otimes 2PBB^T P)e \leq q_3 \|e\|^2 \\ 2\varepsilon_1 \hat{\xi}_2^T (H^{-1} \otimes I_N)e &- e^T (I_N \otimes 2PBB^T P)\hat{\xi}_2 \leq 2q_2 \|e\|\|\hat{\xi}_2\| \end{aligned} \tag{29}$$

Inequality (28) can be rewritten as

$$
\begin{aligned}
\dot{V}_a(t) &\leq -q_1 \|\hat{\xi}_2\|^2 + 2q_2 \|e\| \|\hat{\xi}_2\| + q_3 \|e\|^2 - \varepsilon_2 \delta^T \delta \\
&\leq -q_1 \|\hat{\xi}_2\|^2 + 2q_2 \left( \frac{1}{2\rho} \|e\|^2 + \frac{\rho}{2} \|\hat{\xi}_2\|^2 \right) + q_3 \|e\|^2 - \varepsilon_2 \delta^T \delta \\
&\leq -\sum_{i=1}^{N} \left[ (q_1 - q_2 \rho) \|\hat{\xi}_{i2}\|^2 - \left( \frac{q_2}{\rho} + q_3 \right) \|e_i\|^2 \right] - \varepsilon_2 \delta^T \delta
\end{aligned}
\tag{30}
$$

We have $\dot{V}_a(t) < 0$, and we can obtain

$$
q_1 - q_2 \rho > 0, \quad \|e_i\|^2 < \frac{\gamma_i (q_1 - q_2 \rho)}{q_2 \rho^{-1} + q_3} \|\hat{\xi}_{i2}\|^2
\tag{31}
$$

Then, we have

$$
\dot{V}_a(t) \leq -\alpha V_a(t)
\tag{32}
$$

where $\alpha = \dfrac{\varepsilon_2}{\theta_{\min}(P)}$. By Lyapunov stability theory and Definition 1, one can obtain that all nodes of CCPNs (1) are synchronized under the distributed controller (14) without DoS attacks. The proof is completed. $\square$

The Zeno and singular triggering behaviors are the most common failure behaviors in the ETC strategy, in which the Zeno behavior is defined as multiple transmissions in a continuous time, and singular triggering behavior is defined as no further trigger after a single trigger. The Zeno and singular triggering behaviors can cause the proposed event-triggered strategy to be ineffective, so these two behaviors must be excluded to ensure the efficient transmission of system information. Then, the following Lemmas show that Zeno behavior is excluded and no node exhibits singular triggering behavior.

**Lemma 2.** *For CCPNs (1) without DoS attacks, under the event-triggered controller (14), all nodes will not have Zeno behavior if the inter-event time intervals satisfy*

$$
\tau > \max\{b_1, b_2\}^{-1} \ln \left\{ \sqrt{\frac{\gamma_i (q_1 - q_2 \rho)}{q_2 \rho^{-1} + q_3}} + 1 \right\}
\tag{33}
$$

*where $b_1 = \Gamma_{\max}\{I_N \otimes BK\}$, and*
*$b_2 = \max\{\Gamma_{\max}\{I_N \otimes A + H \otimes BK\} + \Gamma_{\max}\{H \otimes BK + G \otimes \Gamma + I_N \otimes A\}\}$.*

**Proof.** Let $e_i(t) = x_i(t_k^i) - x_i(t)$, then we obtain

$$
\begin{aligned}
\dot{e}_i(t) &= A x_i(t_k^i) - A x_i(t) - \sum_{j=1}^{N} g_{ij} \Gamma x_i(t) - B u_i(t) = A\big(x_i(t_k^i) - x_i(t)\big) \\
&\quad - \sum_{j=1}^{N} g_{ij} \Gamma x_i(t) - BK \sum_{j=1}^{N} a_{ij}^0 \big(x_j(t_k^i) - x_i(t_k^i)\big) - BK b_i^0 \big(x_{N+1}(t) - x_i(t_k^i)\big) \\
&= A\big(x_i(t_k^i) - x_i(t)\big) - \sum_{j=1}^{N} g_{ij} \Gamma x_i(t) - BK \sum_{j=1}^{N} a_{ij}^0 \big(x_j(t) - x_i(t)\big) \\
&\quad - BK \sum_{j=1}^{N} a_{ij}^0 \big(e_j(t) - e_i(t)\big) - BK b_i^0 \big(x_{N+1}(t) - x_i(t)\big) + BK b_i^0 e_i(t)
\end{aligned}
\tag{34}
$$

The system (34) can be rewritten as

$$
\dot{e} = [(I_N \otimes A) - (G \otimes \Gamma) + (H \otimes BK)]e - (H \otimes BK)\delta
\tag{35}
$$

We have

$$
\frac{d}{dt} \left( \frac{\|e_i\|}{\|\hat{\xi}_{i2}\|} \right) \leq \frac{\|\dot{e}_i\|}{\|\hat{\xi}_{i2}\|} + \frac{\|e_i\| \|\dot{\hat{\xi}}_{i2}\|}{\|\hat{\xi}_{i2}\|^2}
\tag{36}
$$

It follows from (36) that

$$\|\dot{e}_i\| \leq \|(I_N \otimes A) + (G \otimes \Gamma) + (H \otimes BK)\| \|e_i\| - \|(H \otimes BK)\| \|\delta_i\| \tag{37}$$

From (36) and (37), it yields

$$
\begin{aligned}
&\frac{d}{dt}\left(\frac{\|e_i\|}{\|\hat{\xi}_{i2}\|}\right) \\
&\leq \frac{\|(I_N \otimes A) + (G \otimes \Gamma) + (H \otimes BK)\| \|e_i\|}{\|\hat{\xi}_{i2}\|} - \frac{\|(H \otimes BK)\| \|\delta_i\|}{\|\hat{\xi}_{i2}\|} + \frac{\|e_i\| \|\dot{\hat{\xi}}_{i2}\|}{\|\hat{\xi}_{i2}\|^2} \\
&\leq \|(I_N \otimes A) + (H \otimes BK)\| \frac{\|e_i\|}{\|\hat{\xi}_{i2}\|} + \|(I_N \otimes BK)\| + (G \otimes \Gamma)\frac{\|e_i\|}{\|\hat{\xi}_{i2}\|} \\
&\quad + \|(H \otimes BK)\| \frac{\|e_i\|}{\|\hat{\xi}_{i2}\|} + (I_N \otimes A)\frac{\|e_i\|}{\|\hat{\xi}_{i2}\|}
\end{aligned}
\tag{38}
$$

Let $y = \dfrac{\|e_i\|}{\|\hat{\xi}_{i2}\|}$, and then we have

$$\dot{y} \leq \max\{b_1, b_2\}(1 + y) \tag{39}$$

It is concluded that $y \leq \phi(t, \phi_0)$, where $\phi(t, \phi_0)$ is the solution of $\dot{\phi} = \max\{b_1, b_2\} * (1 + \phi)$ satisfying $\phi(t, \phi_0) = \phi_0$. From (39), we have

$$\phi(t, \phi_0) = e^{\max\{b_1, b_2\}t} - 1 \tag{40}$$

It yields from (22), that

$$y = \frac{\|e_i\|}{\|\hat{\xi}_{i2}\|} \leq \phi(\tau, 0) = e^{\max\{b_1, b_2\}\tau} - 1 \tag{41}$$

Combining (40) and (41), we can obtain

$$\tau > \max\{b_1, b_2\}^{-1} \ln\left\{\sqrt{\frac{\gamma_i(q_1 - q_2\rho)}{q_2\rho^{-1} + q_3}} + 1\right\} \tag{42}$$

which means that Zeno behavior does not occur. $\quad\square$

**Lemma 3.** *For CCPNs (1) without DoS attacks, under the event-triggered controller (14), no node exhibits singular triggering behavior.*

**Proof.** From (4) and (16), we have

$$
\begin{aligned}
\hat{\xi}_{i2}(t) - \xi_{i2}(t) &= \sum_{j=1}^{N} a_{ij}^0\left(x_j(t_k^i) - x_i(t_k^i)\right) + b_i^0\left(x_{N+1}(t_k^i) - x_i(t_k^i)\right) \\
&\quad - \left\{\sum_{j=1}^{N} a_{ij}^0\left(x_j(t) - x_i(t)\right) + b_i^0\left(x_{N+1}(t) - x_i(t)\right)\right\} \\
&= \sum_{j=1}^{N} a_{ij}^0\left(e_j(t) - e_i(t)\right) - b_i^0 e_i(t)
\end{aligned}
\tag{43}
$$

Then, one can obtain

$$\hat{\xi}_2 - \xi_2 = (-H \otimes I_N)e \tag{44}$$

Combining the event-triggering condition $\|e_i\| \leq \sqrt{\dfrac{\gamma_i(q_1 - q_2\rho)}{q_2\rho^{-1} + q_3}}\|\hat{\xi}_{i2}\|$ and (44), it yields

$$\left| \left\| \hat{\xi}_{i2}\left(t_k^i\right) \right\| - \|\xi_{i2}(t)\| \right| \leq \Gamma_{\max}\{-H \otimes I_N\}\sqrt{\frac{\gamma_i(q_1 - q_2\rho)}{q_2\rho^{-1} + q_3}}\|e_i(t)\| = \sigma c_i\|e_i(t)\| \quad (45)$$

where $c_i = \sqrt{\dfrac{\gamma_i(q_1 - q_2\rho)}{q_2\rho^{-1} + q_3}}$, $\sigma = \Gamma_{\max}\{-H \otimes I_N\}$, $\sigma c_i = \tilde{c}_i$, and

$$\nu_1 = \frac{\left\| \hat{\xi}_{i2}\left(t_k^i\right) \right\|}{1 + \tilde{c}_i} \leq \|\xi_{i2}(t)\| \leq \frac{\left\| \hat{\xi}_{i2}\left(t_k^i\right) \right\|}{1 - \tilde{c}_i} = \nu_2 \quad (46)$$

From (25), we have

$$\begin{aligned}
\xi_{i2}(t) &= \sum_{j=1}^{N} a_{ij}^0\big(x_j(t) - x_i(t)\big) + b_i^0(x_{N+1}(t) - x_i(t)) \\
&= \sum_{j=1}^{N} a_{ij}^0\big(\delta_j(t) - \delta_i(t)\big) - b_i^0\delta_i(t) \\
\Rightarrow \xi_2(t) &= (-H \otimes I_N)\delta \Rightarrow \|\xi_{i2}\| \leq \sigma\|\delta_i\| \leq \sqrt{\frac{V_a(t)}{\theta_{\min}(P)}}
\end{aligned} \quad (47)$$

It is obvious to show that $\|\xi_{i2}(t)\|$ always stays between $\nu_1$ and $\nu_2$ from (46), and $V_a(t)$ strictly decreases to zero from (17) and (32), that is, $\|\xi_{i2}(t)\|$ will eventually decrease to $\nu_1$ based on (46) and (47). The proof is completed. □

**Remark 4.** *Lemmas 2 and 3 present the feasibility of the event-triggered strategy proposed in this paper, which means that every node of the systems is effectively transmitted under this transmission strategy. It is obvious to obtain that $\gamma_i$ can affect the convergence rate from (30). Different from [25,28,30,36], singular triggering behavior of is excluded in Lemma 3. It means that there exists the instant $t_{k+1}^i$ after the current triggering time $t_k^i$ so that at least one event will be triggered. That is, Lemma 3 excludes the singular triggering behavior that there is no transmission attempt for a long time to ensure that the proposed distributed ETC scheme is effective.*

The communication topologies of CCPNs (1) are easily changed by DoS attacks. However, there are few results focusing on switching topologies of CCPNs caused by DoS attacks, which motivates us to discuss this issue.

**Theorem 2.** *For systems (18) with the distributed controller (14) and (15) and the event-triggering condition (22) and (23), if there exists a symmetric matrix $Q$ and $F = B^T Q$ such that the following inequalities are satisfied*

$$-\Psi_{\sigma(t)}\bar{H}_{\sigma(t)} - \left(\bar{H}_{\sigma(t)}\right)^T\Psi_{\sigma(t)} \leq 0 \quad (48)$$

$$0 < \Psi_{\sigma(t)} \leq \beta_1 I_N \quad (49)$$

*and*

$$2\left(QA + A^T Q + \beta_2 Q\Gamma\right) < \beta_3 Q \quad (50)$$

*where $\Psi_{\sigma(t)} \in \{\Psi_1, \Psi_2, \ldots, \Psi_w\}$, $\beta_2 = \dfrac{\theta_{\max}\left(\Psi_{\sigma(t)}G\right)}{\theta_{\min}\left(\Psi_{\sigma(t)}\right)}$, $\beta_3 > 0$, and $\Psi_1, \Psi_2, \ldots, \Psi_w$ are the positive definite matrices, then the following inequality can be obtained:*

$$V_b(t) \leq e^{\beta_3\left(t - t_{2l}^{ij}\right)}V_b\left(t_{2l}^{ij}\right) \quad (51)$$

**Proof.** Constructing the following Lyapunov function

$$V_b(t) = \delta^T \left( \Psi_{\sigma(t)} \otimes Q \right) \delta \tag{52}$$

The time-derivative trajectory of (52) yields

$$
\begin{aligned}
\dot{V}_b(t) &= 2\delta^T \left( \Psi_{\sigma(t)} \otimes Q \right) \left[ \left( I_N \otimes A + G \otimes \Gamma - \bar{H}_{\sigma(t)} \otimes BF \right) \delta - \left( \bar{H}_{\sigma(t)} \otimes BF \right) e \right] \\
&= 2\delta^T \left( \Psi_{\sigma(t)} \otimes Q \right) (I_N \otimes A + G \otimes \Gamma) \delta - 2\delta^T \left( \Psi_{\sigma(t)} \otimes Q \right) \left( \bar{H}_{\sigma(t)} \otimes BF \right) \delta \\
&\quad - 2\delta^T \left( \Psi_{\sigma(t)} \otimes Q \right) \left( \bar{H}_{\sigma(t)} \otimes BF \right) e = 2\delta^T \Psi_{\sigma(t)} \otimes (QA + A^T Q) \delta \\
&\quad + 2\delta^T \left( \Psi_{\sigma(t)} G \otimes Q\Gamma \right) \delta - \delta^T \left( \Psi_{\sigma(t)} \bar{H}_{\sigma(t)} + \left( \bar{H}_{\sigma(t)} \right)^T \Psi_{\sigma(t)} \right) \otimes QBB^T Q \delta \\
&\quad - \delta^T \left( \Psi_{\sigma(t)} \bar{H}_{\sigma(t)} + \left( \bar{H}_{\sigma(t)} \right)^T \Psi_{\sigma(t)} \right) \otimes QBB^T Q e
\end{aligned}
\tag{53}
$$

By Lemma 1 and (53), we can obtain

$$
\begin{aligned}
\dot{V}_b(t) &\le 2\delta^T \Psi_{\sigma(t)} \otimes (QA + A^T Q) \delta + 2\delta^T \Psi_{\sigma(t)} \otimes \left( \frac{\theta_{\max}\left( \Psi_{\sigma(t)} G \right)}{\theta_{\min}\left( \Psi_{\sigma(t)} \right)} \right) Q\Gamma \\
&= 2\delta^T \left\{ \Psi_{\sigma(t)} \otimes \left[ (QA + A^T Q) + \left( \frac{\theta_{\max}\left( \Psi_{\sigma(t)} G \right)}{\theta_{\min}\left( \Psi_{\sigma(t)} \right)} \right) Q\Gamma \right] \right\} \delta \\
&= 2\delta^T \left\{ \Psi_{\sigma(t)} \otimes \left[ (QA + A^T Q) + \beta_2 Q\Gamma \right] \right\} \delta
\end{aligned}
\tag{54}
$$

From the condition of (50) in Theorem 2, (54) is equivalent to

$$\dot{V}_b(t) \le \beta_3 V_b(t) \tag{55}$$

Then, the following inequality holds:

$$V_b(t) \le e^{\beta_3 \left( t - t_{2l}^{ij} \right)} V_b\left( t_{2l}^{ij} \right) \tag{56}$$

The proof is completed. □

**Theorem 3.** *If DoS attacks satisfy Definitions 1 and 2 with $T_0 > 0$, and conditions (14), (17) and (23) hold, then all nodes of CCPNs (1) are synchronized under DoS attacks provided that the following two conditions are satisfied:*
*(1) $\exists\, \theta_1^* \in (0, \beta_3)$, and the frequency of DoS attacks $F_{a_{(t_0,t)}}$ satisfies*

$$F_{a_{(t_0,t)}} = \frac{N_{a_{(t_0,t)}}}{t - t_0} \le \frac{\theta_1^*}{2 \ln(\mu) + (\alpha + \beta_3)\Delta_*} \tag{57}$$

*(2) $\exists\, \tau_a \ge 0$, for $\forall\, T_0 > 0$ such that*

$$\tau_a > \frac{\alpha + \beta_3}{\alpha + \theta_1^*} \tag{58}$$

*where $\beta_3 > 0$, $\mu = \max\{\theta_{\max}(P)/\theta_{\min}(Q),\ \theta_{\max}(Q)/\theta_{\min}(P)\}$.*

**Proof.** Construct the following piecewise Lyapunov function for the CCPNs (1):

$$
V = \begin{cases}
V_b, & t \in \left[ j_m^{ij}, j_m^{ij} + \Delta_m^{ij} + \Lambda_* \right) \\
V_a, & t \in \left[ j_m^{ij} + \Delta_m^{ij} + \Lambda_*, j_{m+1}^{ij} \right)
\end{cases}
\tag{59}
$$

$A_m = \left[ j_m^{ij}, j_m^{ij} + \Delta_m^{ij} + \Lambda_* \right)$ represents the *m*-th attack is activated. For any $t \in A_m$, from (51), one can obtain

$$V(t) \leq e^{\beta_3 \left( t - j_m^{ij} \right)} V_b \left( j_m^{ij} \right) \tag{60}$$

$S_m = \left[ j_m^{ij} + \Delta_m^{ij} + \Lambda_*, j_{m+1}^{ij} \right)$ represents the *m*-th attack is dormant. For any $t \in S_m$, from (32), it follows that

$$V(t) \leq e^{-\alpha \left( t - j_m^{ij} - \Delta_m^{ij} - \Lambda_* \right)} V_a \left( j_m^{ij} + \Delta_m^{ij} + \Lambda_* \right) \tag{61}$$

According to (60), one can obtain

$$V_b^- \left( j_m^{ij} + \Delta_m^{ij} + \Lambda_* \right) \leq e^{\beta_3 \left( \Delta_m^{ij} + \Lambda_* \right)} V_b \left( j_m^{ij} \right) \tag{62}$$

From (60)–(62), then we have

$$
\begin{aligned}
V(t) &\leq e^{-\alpha \left( t - j_m^{ij} - \Delta_m^{ij} - \Lambda_* \right)} e^{\beta_3 \left( \Delta_m^{ij} + \Lambda_* \right)} V_b \left( j_m^{ij} \right) \\
&\leq \mu e^{-\alpha \left| \tilde{\Theta}_{s'} \left( j_m^{ij} + \Delta_m^{ij} + \Lambda_*, t \right) \right|} e^{\beta_3 \left| \tilde{\Theta}_{a'} \left( j_m^{ij}, t \right) \right|} V_b \left( j_m^{ij} \right) \\
&\leq \mu^2 e^{-\alpha \left| \tilde{\Theta}_{s'} \left( j_m^{ij}, t \right) \right|} e^{\beta_3 \left| \tilde{\Theta}_{a'} \left( j_m^{ij}, t \right) \right|} V_a^- \left( j_m^{ij} \right) \\
&\leq \mu^2 e^{-\alpha \left| \tilde{\Theta}_{s'} \left( j_{m-1}^{ij}, t \right) \right|} e^{\beta_3 \left| \tilde{\Theta}_{a'} \left( j_m^{ij}, t \right) \right|} V_a^- \left( j_{m-1}^{ij} + \Delta_{m-1}^{ij} + \Lambda_* \right) \\
&\quad \cdots \\
&\leq \mu^{2 N_a{}_{(t_0,t)}} e^{-\alpha \left| \tilde{\Theta}_{s'} (t_0,t) \right|} e^{\beta_3 \left| \tilde{\Theta}_{a'} (t_0,t) \right|} V(t_0)
\end{aligned}
\tag{63}
$$

For all $t > t_0$, $\left| \tilde{\Theta}_{a'}(t_0,t) \right| \leq \left| \tilde{\Theta}_a(t_0,t) \right| \leq \left[ \left| \Theta_a(t_0,t) \right| + \left( 1 + N_{a_{(t_0,t)}} \right) \Lambda_* \right]$, and $\left| \tilde{\Theta}_{s'}(t_0,t) \right| = t - t_0 - \left| \tilde{\Theta}_{a'}(t_0,t) \right|$. Therefore, we have

$$
\begin{aligned}
&-\alpha \left( t - t_0 - \left| \tilde{\Theta}_{a'}(t_0,t) \right| \right) + \beta_3 \left| \tilde{\Theta}_{a'}(t_0,t) \right| \\
&= -\alpha(t - t_0) + (\alpha + \beta_3) \left[ \left| \Theta_a(t_0,t) \right| + \left( 1 + N_{a_{(t_0,t)}} \right) \Lambda_* \right]
\end{aligned}
\tag{64}
$$

Combining (63) and (64), we have

$$
\begin{aligned}
V(t) &\leq \mu^{2 N_a{}_{(t_0,t)}} e^{-\alpha \left| \tilde{\Theta}_{s'} (t_0,t) \right|} e^{\beta_3 \left| \tilde{\Theta}_{a'} (t_0,t) \right|} V(t_0) \\
&\leq e^{(\alpha + \beta_3)(T_0 + \Lambda_*)} e^{-\alpha(t - t_0)} e^{\frac{(\alpha + \beta_3)}{\tau_a}(t - t_0)} e^{[2 \ln \mu + (\alpha + \beta_3) \Lambda_*] N_a{}_{(t_0,t)}} V(t_0)
\end{aligned}
\tag{65}
$$

From (57), (58) and (60) can be rewritten as

$$V(t) \leq e^{(\alpha + \beta_3)(T_0 + \Lambda_*)} e^{-\theta_1(t - t_0)} V(t_0) \tag{66}$$

where $\theta_1 = \alpha - \frac{(\alpha + \beta_3)}{\tau_a} - \theta_1^* > 0$. This is equivalent to all nodes of CCPNs (1) being synchronized in the presence of DoS attacks. The proof is completed. $\square$

**Remark 5.** *The synchronization problem was investigated on an assumption that each node maintains communication connection with other nodes in [37]. On the contrary, these assumptions can be removed in this paper, and the triggering time of each node is not affected by each other. In addition, the frequency and duration of DoS attacks are accurately characterized in Theorem 3. In this respect, the designed parameter $\mu$ needs to satisfy the fixed constraints to ensure that $\mu^{N_a{}_{(t_0,t)}}$ is converged.*

## 4. Numerical Examples

**Example 1.** *Consider the CCPNs as shown in Figure 2, which have five nodes (node 5 is an isolated node). The parameters are chosen as follows:*

$$A = \begin{bmatrix} -5 & 4 & 7 \\ -3 & -7 & -4 \\ -2 & -7 & -9 \end{bmatrix}, G = \begin{bmatrix} 1 & -1 & 0 & 0 \\ 0 & 1 & -1 & 0 \\ 0 & 0 & 1 & -1 \\ -1 & 0 & 0 & 1 \end{bmatrix}, B = \begin{bmatrix} 0.4 \\ 1.7 \\ 0.3 \end{bmatrix},$$

$\Gamma = diag(0.1, 0.1, 0.1), \varepsilon_{\min} = 7.236, \varepsilon_1 = 1, \gamma_i = 0.5, q_1 = 6.8541, q_2 = 5.2361, q_3 = 2.7058, \varepsilon_2 = 6.236, \alpha = 19.9552.$

*When the complex cyber–physical networks are not attacked by DoS attacks, one can obtain the H matrix by the communication topologies as shown in Figure 2.*

$$H = \begin{bmatrix} 3 & -1 & 0 & -1 \\ -1 & 1 & 0 & 0 \\ 0 & -1 & 2 & -1 \\ 0 & 0 & -1 & 2 \end{bmatrix}.$$

*By Theorem 1, we obtain the controller gain K and Lyapunov variable P as follows:*

$$K = \begin{bmatrix} -2.4163 & 8.8224 & -4.9069 \end{bmatrix},$$

$$P = \begin{bmatrix} 1.0221 & -0.5332 & 0.5421 \\ -0.5332 & 0.9725 & -0.7241 \\ 0.5421 & -0.7241 & 1.1136 \end{bmatrix}.$$

*The initial states of nodes are $x_1 = \begin{bmatrix} -2 & 0.5 & 0.5 \end{bmatrix}^T$, $x_2 = \begin{bmatrix} 2 & -3 & 4 \end{bmatrix}^T$, $x_3 = \begin{bmatrix} 2 & 3 & 3 \end{bmatrix}^T$, $x_4 = \begin{bmatrix} -3.5 & -0.5 & 1 \end{bmatrix}^T$.*

*As shown in Figure 3, the trajectories of error states between follower nodes and the isolated node approach zero when the DoS attacks are dormant. Figures 4 and 5 show the trajectories of synchronization errors based on the ETC strategy and transmission errors of nodes, which prove that all nodes of the systems are synchronized with the distributed controller. The transmission attempts of nodes based on the ETC strategy are shown in Figure 6. It means that each node's triggering rules are independent. It can be seen from Figures 3, 4 and 6 that the proposed ETC strategy saves communication resources and reduces the unnecessary transmission of the systems while ensuring the convergence effect. The above simulation results show that when the external attacks are zero, the system realizes the synchronization of all nodes based on the distributed event-triggered controller, which verifies the validity of Theorem 1 in this paper.*

*When the systems are attacked by the DoS attacks, it is supposed that two edges can be attacked separately or simultaneously. Figure 7b–d show several network topologies generated by DoS attacks. Figure 8 shows the DoS signal, where mode 0 denotes DoS dormancy and mode 1 denotes DoS activation. Suppose that three topologies caused by DoS attacks are shown as in Figure 8. We have*

$$\overline{H_1} = \begin{bmatrix} 2 & -1 & 0 & 0 \\ -1 & 1 & 0 & 0 \\ 0 & 0 & 1 & -1 \\ 0 & 0 & 0 & 1 \end{bmatrix}, \overline{H_2} = \begin{bmatrix} 2 & -1 & 0 & 0 \\ -1 & 1 & 0 & 0 \\ 0 & -1 & 2 & -1 \\ 0 & 0 & -1 & 2 \end{bmatrix},$$

$$\overline{H_3} = \begin{bmatrix} 3 & -1 & 0 & -1 \\ -1 & 1 & 0 & 0 \\ 0 & 0 & 1 & -1 \\ 0 & 0 & -1 & 2 \end{bmatrix}.$$

The parameters are chosen as $\beta_1 = 15, \beta_2 = 0.3843, \beta_3 = 1$. By Theorem 2, we obtain the controller gain F and Lyapunov variable Q as follows:

$$F = \begin{bmatrix} -0.7125 & 6.5357 & -5.1397 \end{bmatrix},$$

$$Q = \begin{bmatrix} 0.7845 & -0.6392 & 0.9091 \\ -0.6392 & 2.8395 & -2.7592 \\ 0.9091 & -2.7592 & 3.4799 \end{bmatrix}.$$

Then, we obtain $\theta_{\max}(P) = 2.2434, \theta_{\min}(P) = 0.3125, \theta_{\max}(Q) = 6.1641, \theta_{\min}(Q) = 0.3172, \mu = 19.7251$. Let $\theta_1^* = 0.8448$, then, one can obtain

$$F_{a_{(t_0,t)}} = \frac{N_{a(t_0,t)}}{t - t_0} \leq \frac{\theta_1^*}{2\ln(\mu) + (\alpha + \beta_3)\Lambda_*} = 0.316, \tau_a > \frac{\alpha + \beta_3}{\alpha + \theta_1^*} = 1.001.$$

This means that the DoS attacks cannot occur more than 0.316 s during a unit of time and duration smaller than 1.001 s. Figure 9 shows the trajectories of error states between follower nodes and the isolated node converge to zero under DoS attacks. Figures 10 and 11 show the trajectories of synchronization errors based on the ETC strategy and transmission errors of nodes, which prove that all nodes of the systems are synchronized with the distributed controller. It can be seen from Figures 9 and 10 that the proposed ETC strategy saves communication resources and reduces unnecessary transmission of the systems while ensuring the convergence effect. The above simulation results show that as long as the dwell time and attack frequency of DoS attacks satisfy the upper limit constraints calculated in this paper, the complex cyber–physical networks can achieve the synchronization of all nodes with the designed ETC strategy, which also verifies the effectiveness of Theorem 3 to calculate the tolerable attacks.

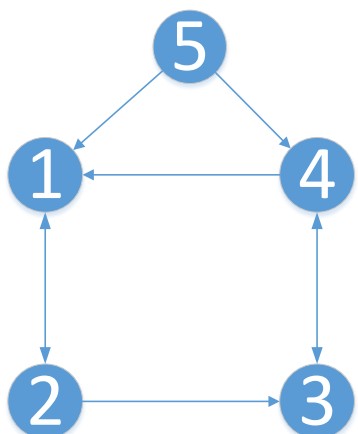

**Figure 2.** Fixed topology.

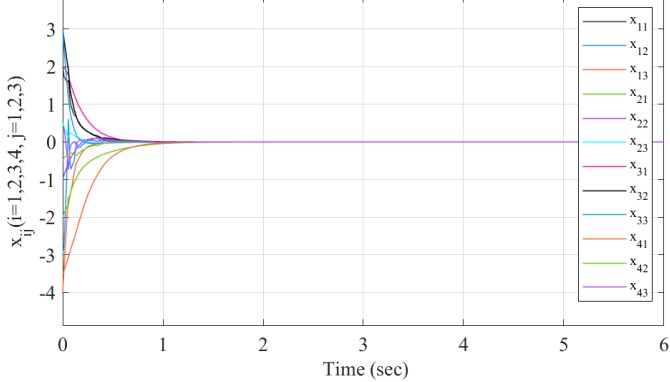

**Figure 3.** Tracking error trajectories of nodes without DoS attacks.

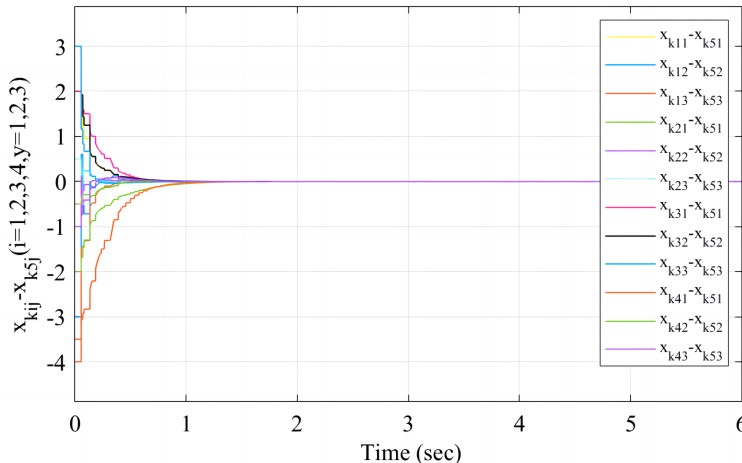

**Figure 4.** Tracking error trajectories of nodes with event-triggered protocol without DoS attacks.

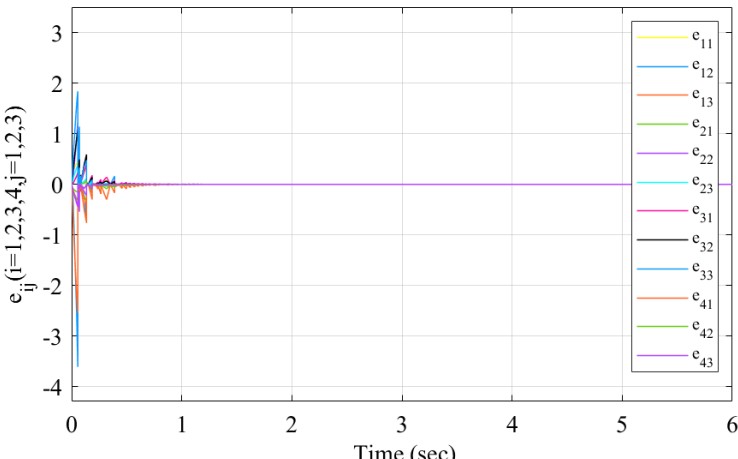

**Figure 5.** Error trajectories of nodes without DoS attacks.

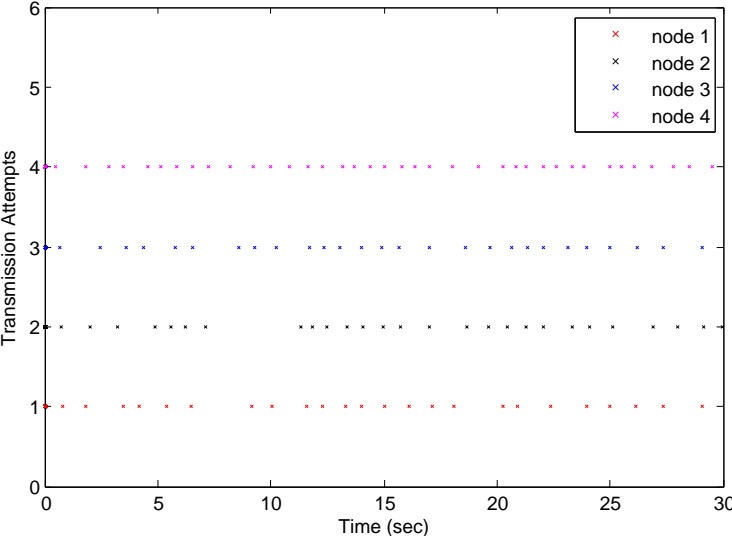

**Figure 6.** Transmission attempts of follower nodes.

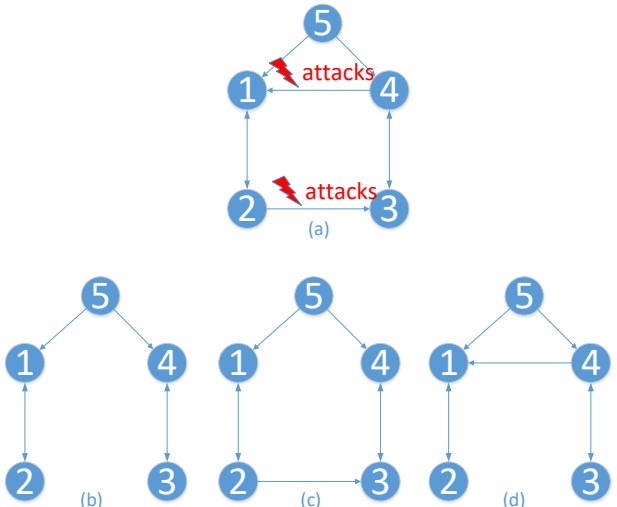

**Figure 7.** Switching topologies.

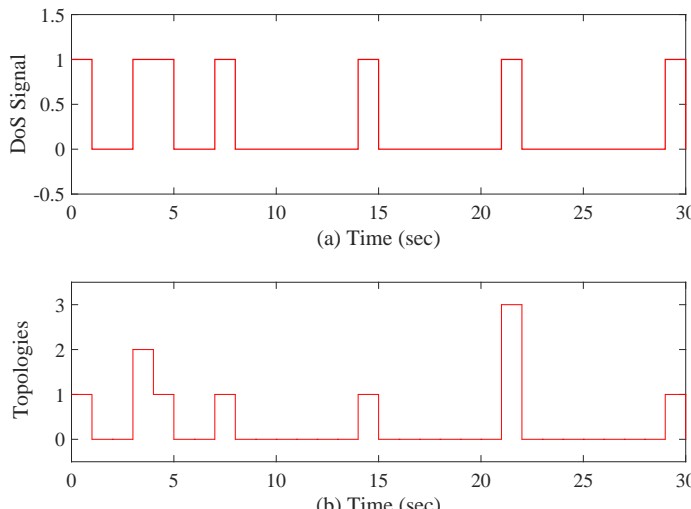

**Figure 8.** DoS signal and switching signal.

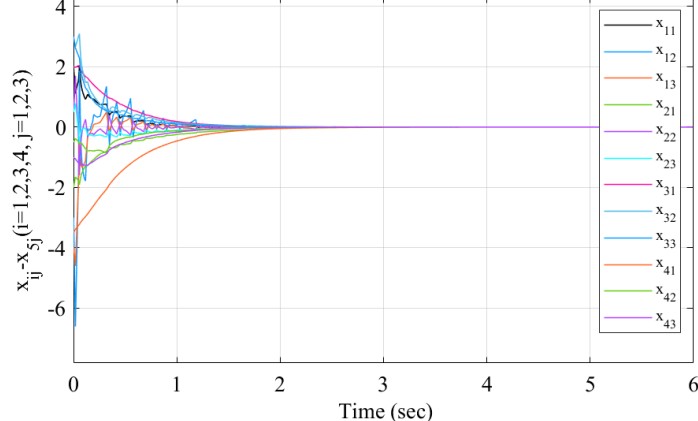

**Figure 9.** Tracking error trajectories of nodes under DoS attacks.

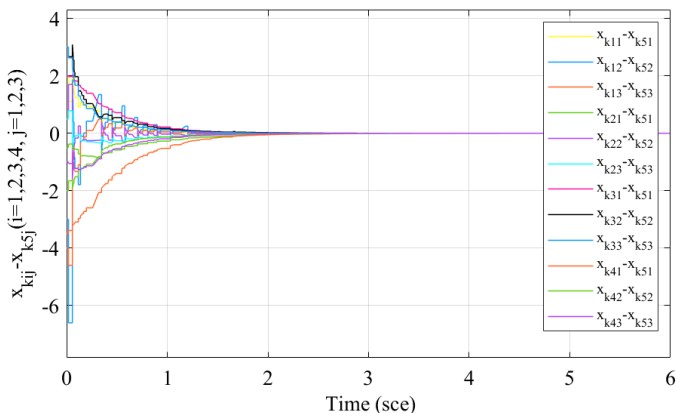

**Figure 10.** Tracking error trajectories of nodes with event-triggered protocol under DoS attacks.

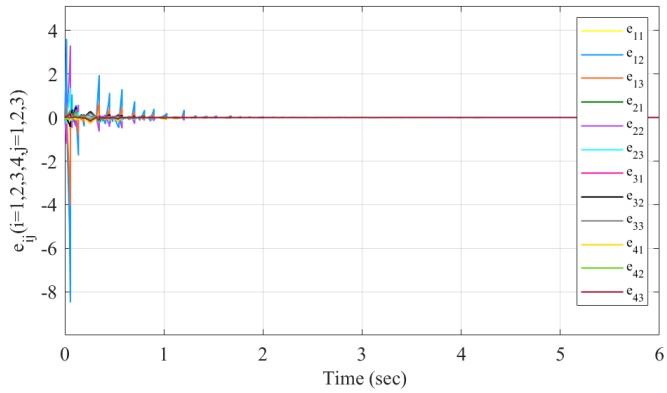

**Figure 11.** Error trajectories of nodes under DoS attacks.

**Example 2.** *In order to verify the superiority of the proposed method, we consider the security control problem solved in [38]. The parameters borrowed from [38] are as follows:*

$$A = \begin{bmatrix} 0 & -1 \\ 2 & 1 \end{bmatrix}, B = \begin{bmatrix} 1 & 0 \\ 0 & 2 \end{bmatrix}, H = \begin{bmatrix} 3 & -1 & -1 \\ -1 & 3 & -1 \\ -1 & -1 & 3 \end{bmatrix},$$

$$\overline{H_1} = \begin{bmatrix} 2 & -1 & 0 \\ -1 & 3 & -1 \\ 0 & -1 & 2 \end{bmatrix}, \overline{H_2} = \begin{bmatrix} 1 & 0 & 0 \\ 0 & 2 & -1 \\ 0 & -1 & 2 \end{bmatrix}.$$

*The DoS attack behavior and topologies are the same as [38], as shown in Figures 12 and 13. Other parameters are chosen as follows:*

$$\varepsilon_{\min} = 5.231, \varepsilon_1 = 1, \gamma_{\min} = 0.43, q_1 = 4.783, q_2 = 6.893,$$
$$q_3 = 2.785, \alpha = 7.981, \beta_1 = 13, \beta_2 = 0.2983, \theta_1^* = 0.5897, \beta_3 = 18.$$

*By Theorems 1 and 3, one can obtain the controller gains K and F as follows:*

$$K = \begin{bmatrix} -1.6521 & 0.7893 \\ 0.0672 & -2.0972 \end{bmatrix}, F = \begin{bmatrix} -1.7624 & -0.9821 \\ 0.0762 & -0.9879 \end{bmatrix}.$$

*and*

$$F_{a_{(t_0,t)}} = \frac{N_{a(t_0,t)}}{t - t_0} \leq \frac{\theta_1^*}{2\ln(\mu) + (\alpha + \beta_3)\Lambda_*} = 0.1087, \tau_a > \frac{\alpha + \beta_3}{\alpha + \theta_1^*} = 4.1253.$$

*This means that the DoS attacks cannot occur more than 0.1087 s during a unit of time and duration smaller than 4.1253 s. The initial values of the error systems borrowed from [38] is* $e(0) = \begin{bmatrix} 0.25 & 0.09 & 0.09 \end{bmatrix}^T$.

*Figure 14 shows the trajectories of synchronization errors based on the ETC strategy. It is easy to see that all nodes of the systems are synchronized under the distributed controller after 110 s compared with 600 s in [38], which verifies the superiority of the results. The above simulation results show that as long as the dwell time and attack frequency of DoS attacks satisfy the upper limit constraints calculated in this paper, the complex cyber–physical networks can achieve the synchronization of all nodes by the designed ETC strategy.*

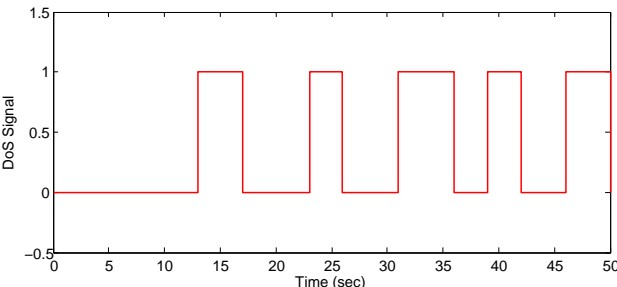

**Figure 12.** DoS attack signal.

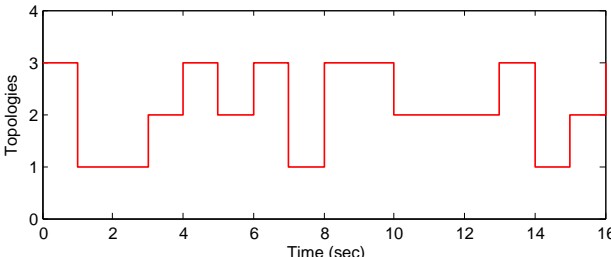

**Figure 13.** Topologies of systems.

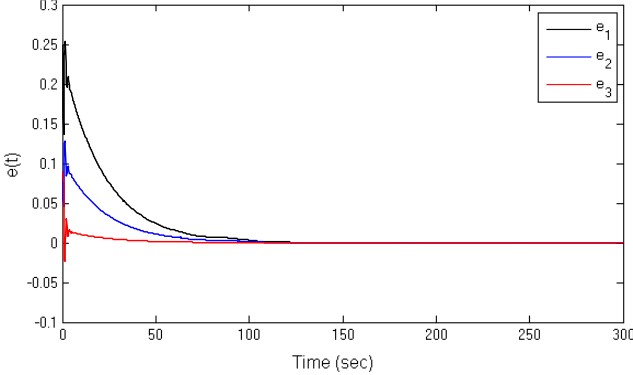

**Figure 14.** Error trajectories of nodes under DoS attacks.

## 5. Conclusions

This paper addressed the distributed event-triggered synchronization problem of CCPNs in the presence of DoS attacks. A distributed event-triggered controller using two combined measurements was designed to achieve synchronization and save the limited communication resources. In addition, the Zeno and singular triggering behaviors of the proposed event triggering strategy were excluded to ensure the effectiveness of states transmission. Since DoS attacks can generate many communication topologies, sufficient conditions for synchronization under finite energy attack are obtained by using a piecewise

Lyapunov function. We will focus on the problem of attack detection and resilient control of CCPNs under mixed attacks in future work.

**Author Contributions:** Methodology, X.H. and D.-W.D.; validation, Y.X. and D.-W.D.; Writing—original draft preparation, X.H.; Writing—review and editing, Y.X. All authors have read and agreed to the published version of the manuscript.

**Funding:** This research was funded by the Postdoctoral Research Foundation of Shunde Innovation School, University of Science and Technology Beijing under Grant 2021BH010.

**Institutional Review Board Statement:** Not applicable.

**Informed Consent Statement:** Not applicable.

**Data Availability Statement:** Not applicable.

**Conflicts of Interest:** The authors declare no conflict of interest.

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
