# Peer review of "Distributed Event-Triggered Synchronization for Complex Cyber–Physical Networks under DoS Attacks"

_applsci, doi:10.3390/app13031716_

Round 1
Reviewer 1 Report
There are many grammatical errors in the text, which make understanding the presented content almost impossible. There are very few attempts to elaborate on defined terms. Not enough motivation and background information is provided.
- What has been done on this topic, and how is your work different than others? You have just given a single sentence with citations to other related works without elaborating on them and guiding us on how your work is different.
- Give us a real-world example of an "Event-triggered control-based multi-agent system." You just defined the term without giving any examples.
- What type of DoS attack are you investigating? Is it the application layer? Is it a network bandwidth attack? You have just used a blanket statement of DoS. Provide a clear "threat model" what are your assumptions of the threat actors?
- There are many terms that are used without defining them first, e.g., what's "Zeno" behaviour?
Reviewer 2 Report
The manuscript addresses the distributed event-triggered synchronization problem of CCPNs in the presence of DoS attacks.
- Which is the motivation (abstract)
- CCPN, ETC not defined in introduction
- Square out of context after "The proof is completed" sentence. What is the purpose of those squares?
- What do you mean by isolated in node 5 from Figure 2
- How did you choose the initial states of nodes?
- Missing y axis text in Fig3-6
All figures require a deep description and conclusion. There is not a deep state of the art study. You should compare your work with a baseline.
Round 2
Reviewer 1 Report
The changes have addressed my comments.
Reviewer 2 Report
The manuscript has improved a little bit, however, it requires some changes wrt the results and simulation setup. The y-axis from fig 3-4 are not clear. All the plots require a deeper analysis. You say that the initial config was random. How you expect to replicate the simulation if you cannot control the initial parameters? You need to justify the setup for reproducibility and comparison with other configuration.
